# Influence of Curing Temperature on the Performance of Calcined Coal Gangue–Limestone Blended Cements

**DOI:** 10.3390/ma17081721

**Published:** 2024-04-09

**Authors:** Wenguang Zhang, Hao Zhou, Yueyang Hu, Jiaqing Wang, Jian Ma, Ruiyu Jiang, Jinfeng Sun

**Affiliations:** 1Key Laboratory for Advanced Technology in Environmental Protection of Jiangsu Province, Yancheng Institute of Technology, Yancheng 224051, China; wenguangzhang1997@163.com (W.Z.); jiangry@ycit.edu.cn (R.J.); 2Nanjing Institute of Environmental Sciences, Ministry of Ecology and Environment of the People’s Republic of China, Nanjing 210042, China; zhouhao@nies.org; 3College of Materials Science and Engineering, Yancheng Institute of Technology, Yancheng 224051, China; huyueyang1989@163.com; 4College of Civil Engineering, Nanjing Forestry University, Nanjing 210037, China; jiaqingw@njfu.edu.cn

**Keywords:** calcined coal gangue, limestone, curing temperature, hydration, microstructure, compressive strength

## Abstract

The utilization of calcined coal gangue (CCG) and limestone for the preparation of blended cement is an efficient approach to address the issue of coal gangue disposal. However, the compressive strength development of blended cement is slow, particularly at high substitution levels of CCG. Therefore, this study aimed to promote the hydration and mechanical properties of the calcined coal gangue–limestone blended cements by increasing the curing temperature. In this study, the samples were cured at two different temperatures, namely 20 and 40 °C. The four groups of samples contained 15 wt.%, 30 wt.%, 45 wt.% and 60 wt.% cement substitutions using CCG and limestone (2:1 mass ratio). The compressive strength, hydration and microstructure were investigated at the ages of 1 to 28 d. X-ray diffraction (XRD) and thermogravimetry (TG) were used to study the hydration behavior of samples. Mercury intrusion porosimetry (MIP) and scanning electron microscopy (SEM) were used to determine the microstructure of the samples. The results indicate that an increase in curing temperature significantly promotes the compressive strength of the calcined coal gangue–limestone blended cements from 1 to 28 d. The microstructural analysis indicates that increasing the curing temperature not only promotes cement hydration but also facilitates the reaction of CCG, which precipitated more hydrates such as C-A-S-H gel, Hc and Mc. These hydrates are conducive to refining the pore structures and densifying the microstructure, which sufficiently explains the enhanced compressive strength of the calcined coal gangue–limestone blended cements.

## 1. Introduction

Coal is an important energy resource in many countries. The top five countries holding proved coal reserves globally are the United States (22%), Russia (15%), Australia (14%), China (14%), and India (11%) [1]. Coal gangue (CG) is one of the largest industrial wastes, which is mainly discharged in the process of coal mining and washing. China has large reserves of about 6 billion tons of CG and is expected to increase by approximately 500–800 million tons for each year [2]. However, its utilization rate is less than 20%. CG is mainly composed of SiO_2_ and Al_2_O_3_ (50–70% wt.%), even though its chemical compositions vary with different sources [3]. Such high SiO_2_ and Al_2_O_3_ contents make it possible to be used as a potential SCM in cement-based materials [4]. The current research has demonstrated that CG can be used as a potentially active SCM to partially replace cement after calcination at appropriate temperatures (usually 700–900 °C) [5]. In the calcination process, kaolinite (Al_2_O_3_·2SiO_2_·H_2_O, AS_2_H_2_) would dehydroxylate to form metakaolin (Al_2_O_3_·2SiO_2_, AS_2_) [6,7], which was the main source of the pozzolanic reactivity of calcined coal gangue (CCG) [8].

Extensive studies indicate incorporating an appropriate amount of CCG as a partial replacement of cement can maintain a relatively high long-term mechanical strength [9,10]. This is because as the curing time increased, the amorphous metakaolin with high pozzolanic reactivity in CCG gradually started to react with portlandite (CH) to form additional C-A-S-H gel [11], which resulted in a higher strength growth rate as compared with the pure cement especially from 7 days onwards [12]. In this process, pore structure distributions in the hydrated binder have also been refined [13], which is reflected in the reduction in the large pores and formation of more gel pores with lower size. More importantly, the pore structure refinement and ion consolidation ability of C-A-S-H gel are also favorable for resisting the penetration and transmission of corrosive ions, such as Cl^−^ and SO_4_^2−^ [14]. Therefore, CCG-blended cement also secure satisfactory long-term durabilities in severe environments as compared with the corresponding reference samples. Additionally, the pozzolanic reactivity [15], fineness [16], and replacement level of CCG were also found to influence the performance of the CCG-blended cement. The application of CCG is generally limited to 10–20 wt.%. However, with a higher replacement amount, the performance of composites cement would decrease remarkably.

In recent years, some researchers have begun to introduce limestone to the binary system of calcined coal gangue-cement [17,18], which is originated from limestone calcined clay cement (LC^3^) [19]. In LC^3^, the high pozzolanic reactivity of calcined kaolinite clay (metakaolin) and the synergistic effect between metakaolin and limestone produce C-A-S-H gel, hemicarboaluminate (Hc) and monocarboaluminate (Mc) [20], which allows a higher cement replacement of up to 50 wt.% in LC^3^ without comprising its compressive strength [21]. Given that kaolinite is also one of the major minerals in CG, it is theoretically feasible to use CCG instead of calcined kaolinite clay in LC^3^. Liu et al. [18] reported that the co-utilization of CCG (with kaolinite content of 28.4%) and limestone maximized the level of substitution of cement up to 30% (2:1 mass ratio). The ternary binder comprising 10% CCG and 5% limestone achieved comparable compressive strength to the reference mortar at 7 d. Moreover, for a given replacement level, the limestone can enhance the compressive strength of CCG blended cement by approximately 5–12% from 7 d onwards. The authors [18] attribute this observation to the synergistic effect between CCG and limestone, forming more hydrates (C-A-S-H gel, Hc and Mc) and refining the pore structure. As shown by Jiu et al. [17], compared with the reference cement paste, the paste with 40–50% partial replacement of cement by combined CCG and limestone (1:1, 2:1 and 3:1 mass ratio) exhibited a 14–35% increase in the 28-day compressive.

The limestone calcined coal gangue composite provides satisfactory late-strength advantages. However, the existing research also concluded that both calcined coal gangue-blended cements and the limestone-calcined coal gangue composite present characteristic of much slower compressive strength development compared to the reference Portland cement, especially at higher replacement levels. Take the work of Liu et al. [18] as an example: the mortar prepared with 20% CCG and 10% limestone showed a 46.6% decrease in the 1-day compressive strength compared with the reference mortar. This is also similar to other SCM-blended cements [22,23]. Most of these observations can be related to the significant lower reaction kinetics of SCMs than that of clinker phases [24]. In practical engineering, the slow early-strength development usually limits the large-scale utilization of SCMs.

It has been reported that approximately increasing the curing temperature is beneficial for achieving a higher early compressive strength of cement-based materials [25,26]. When the curing temperature is increased, the hydration kinetics of cement is greatly enhanced, which is reflected in the increased cement hydration degree [25] and is also supported by the classical boundary nucleation and growth (BNG) model [27,28]. More hydrates such as C-S-H gel are precipitated and effectively contributed to the early compressive strength development of pure cement mortar [29]. As for the SCMs-blended cement, especially a higher 30% substitution of SCMs for cement [30,31], these promotional effects are more significant. Many of the results indicate that the reactions of SCMs were also enhanced apart from cement hydration [32,33], which were also the main contributors to the compressive strength development at early and later curing ages. However, no relevant studies have reported the effect of curing temperature on the performance of calcined coal gangue–limestone blended cements so far. This part of the study should be paid attention to, because if the early strength of this blended cement can be improved by increasing the curing temperature, the application of calcined coal gangue in cement-based cementitious materials can be greatly promoted. In addition, the resource treatment of coal gangue can be accelerated to a certain extent.

Based on the research gaps of the existing studies, this research aimed at investigating the effect of curing temperature on hydration and the microstructure properties of calcined coal gangue–limestone blended cements. Four groups of samples with 15 wt.%, 30 wt.%, 45 wt.% and 60 wt.% cement substitutions using calcined coal gangue and limestone (2:1 mass ratio) were designed for the experimental investigations. The curing temperatures were set at 20 and 40 °C up to 28 d. These temperature are usually set as the curing temperature in the current research, which can cover ranges representative of common practice in a place of perennial heat [33]. The evolution in the compressive strength of the mortars was first examined. Then, the X-ray diffraction (XRD), thermogravimetry (TG), mercury intrusion porosimetry (MIP) and scanning electron microscopy (SEM) tests were performed on the corresponding pastes to disclose the mechanism of compressive strength development.

## 2. Materials and Methods

### 2.1. Starting Materials

The Portland cement (P.I. 42.5) used in this study was obtained from Fushun Cement Co., Ltd. (Fushun, China) Limestone with a purity of 85% and was provided by Zaozhuang Building Materials Co., Ltd. (Zaozhuang, China). The coal gangue was collected from Huainan City, Anhui Province. Table 1 summarizes the chemical composition of each material. After being received, the coal gangue was further dried, crushed, ground and finally calcined at 800 °C for 2 h. Figure 1a shows the XRD patterns of the coal gangue before and after calcination. As observed, mica, kaolinite, quartz, and siderite are the major minerals in CG. After calcination, the characteristic peaks for kaolinite (2θ = 12.29°, 24.85°, 28.75°, 34.96°, 37.73°, 62.18°) disappeared by calcining at 800 °C, which clearly indicates a complete decomposition of kaolinite. It can be calculated from TG curves (Figure 1b) using the tangent method that the kaolinite content of coal gangue was approximately 14.1%. Additionally, the sand complied with GB/T 17671-2021 [34] and provided by Xiamen ISO Standard Sand Co., Ltd. (Xiamen, China) was used for the mortars’ preparation.

### 2.2. Mixture Proportions

The detailed mixture proportions of mortars are displayed in Table 2. Mortars were prepared with a constant water to binder (w/b, binder = cement + limestone + calcined coal gangue) ratio of 0.5 and a constant sand to binder ratio of 3. The pure Portland cement mortar was used as the reference sample. Four different limestone and calcined coal gangue combinations were designed, and they were, respectively, at total OPC replacement levels of 15%, 30%, 45% and 60% by mass. The mass ratio between calcined coal gangue and limestone was fixed at 2:1. Specially, additional gypsum was added to achieve an adequate sulfate balance of the systems. As shown in Table 2, the notation was given according to the curing temperatures, constituents and the corresponding weight percentage in the binders. Take T40-CCGL60 as an example; it refers to the sample consisting of 40 wt.% CCG and 10 wt.% limestone, which was cured at the temperature of 40 °C.

### 2.3. Mortars and Pastes Preparation

They were prepared according to the Chinese standard GB/T 50081-2019 [35]. All materials involved in Table 2 were respectively pre-cured at the target temperatures for 1 d. Cement, CCG, limestone and gypsum were weighed according to Table 2, and they were uniformly mixed using a high-shear mixer. After this, the above mixtures, sand and water (or PCE if needed) were mixed to prepare the mortar. The fresh mortar was cast into cubic samples with a size of 40 × 40 × 40 mm^3^ followed by being, respectively, cured at the temperatures of 20 and 40 °C for 1 d. Then, the samples were removed from the molds and transferred to a water curing tank with the same temperatures as that of the fresh mortar for the compressive strength test at the designated curing ages.

Pastes were used for better revealing the mechanism of compressive strength changes resulted from altering the curing temperatures. Their mix proportions were referred to Table 2 without taking sand and PCE into consideration. The procedure of paste preparations and the curing conditions are the same as that of the mortar. Once at the required testing ages, the samples were crushed into small pieces and then were immersed in ethanol for 7 d to stop the hydration. Afterwards, the ethanol was decanted, and part of the piece samples was directly dried in a vacuum drying oven at 40 °C for 48 h. Finally, these samples were used for the MIP and SEM analysis. Another part of the pieces without being dried was finely ground to <75 μm, which was followed by the same drying regime in the vacuum-drying oven for XRD and TG tests. It is notable that all hydration-stopped samples were always kept in the vacuum-drying oven to avoid being carbonated until the tests.

### 2.4. Analytical Techniques

The compressive strength test followed GB/T 50081-2019, and it was performed on a YYW-300 DS hydraulic testing machine (Zhejiang Yiyu Instrument Equipment Co., Ltd., Shaoxing, China) with a 300 kN capacity. Mortars were tested at the required curing ages of 1 day, 3, 7 and 28 days. The applied loading rate on the samples was 2400 N/s. For each group of the sample, the average of the three tests is considered as the final compressive strength.

A Rigaku SmartLab 3000A X-ray diffractometer (Rigaku, Tokyo, Japan) equipped with CuKα X-rays was used to determine the mineralogical composition of the hydrated pastes. The 2θ scan range was 5–65° with a scan speed of 5°/min, and the operating conditions were 15 KV and 40 mA.

A NETZSCH STA 449F3 differential thermal analyzer (NETZSCH Analyzing & Testing, Selb, Germany) was used for the TG-DTG analysis of the hydrated paste samples with samples subjected to a temperature range of 40 °C to 1000 °C and a temperature increase rate of 10 °C/min.

A PoreMaster GT produced by Quantachrome (Boynton Beach, FL, USA) is used for analysis of the pore structure of the paste samples. The intrusion pressure was gradually increased from 0.10 to 60 psia, and the pores ranging from 3.6 nm to 950 μm can be probed.

A Nova NanoSEM 450 scanning electron microscope (FEI, Hillsboro, OR, USA) was used to observe the morphology of the hydrated paste samples. All samples were coated with a layer of gold to increase the conductivity of samples.

## 3. Results and Discussion

### 3.1. Compressive Strength

Figure 2 displays the compressive strength of the mortars cured at 20 and 40 °C up to 28 d. As shown in Figure 2a, the compressive strength of mortars at 1 d decreased with the increase in the replacement level of cement regardless of the curing temperature. This is expected and can be explained by the dilution of cement with an increased content of limestone and CCG and a much lower reaction activity of limestone and CCG than that of cement, which resulted in a significant reduction in the compressive strength. On the contrary, it can be clearly seen that the increased curing temperature evidently promoted the 1-day compressive strength. For the reference mortar, its 1-day compressive strength achieved 13.1 MPa at 20 °C, while this value increased to 19.4 MPa at 40 °C with an increase of about 48.7%. Similar trends can also be observed in the mortars with adding CCG and limestone, but their strength development tends to be more sensitive to curing temperature, especially for higher coupled limestone and CCG content. Take CCGL15 as an example, the curing temperature of 40 °C resulted in an increase in the compressive strength by approximately 51% as compared to 20 °C. Similarly, for CCGL15, the corresponding growth rate increased up to about 420%.

Similarly, the strength development of the mortars at 3 (Figure 2b) and 7 d (Figure 2c) also displays a close dependence on the curing temperatures. In the case of the reference mortar, the compressive strength at 3 d increased by 5.1 MPa from 20 to 40 °C, but it was comparable at 7 d. Studies have shown that increasing the curing temperature could accelerate the early age of cement hydration and promote early strength development [26]. Unlike the reference mortar, the increase in curing temperature from 20 to 40 °C enhanced the compressive strength of CCGL15, CCGL30, CCGL45 and CCGL60, respectively, by 3.5, 6.3, 7.6 and 8.3 MPa at 3 d, and 3.5, 6.3, 7.6 and 8.3 MPa at 28 d. These faster strength gains than the reference mortar could be possibly interpreted as there being faster reactions and more hydrates generated in these CCG- and limestone-added systems as the consequence of increased curing temperature. Additionally, it is also interesting to note that the CCGL30 mortars gained comparable compressive strength to the CCGL15 mortars with a curing temperature of 40 °C both at 3 and 7 d, indicating that the compressive strength of the lower clinker system is more sensitive to the curing temperature.

With the curing age increased up to 28 d (Figure 2d), the curing temperature displays limited influence on the compressive strength of reference mortar, which is reflected by the comparable strength values upon increasing the curing temperature from 20 to 40 °C. On the contrary, for different CCGL mortars, their strength values at 40 °C are expected to be higher than at 20 °C, indicating that increasing the curing temperature is still beneficial to developing the compressive strength of CCGL mortars. But the difference in the strength values between 20 and 40 °C appears smaller and smaller, which is possibly related to the great reaction of CCG in this period that makes up for the strength development of mortars with CCG and limestone to some extent.

In order to better disclose the potential mechanism of the compressive strength development of these mortars, the systems of OPC and CCGL30 curing at 20 and 40 °C were selected for the further microstructural analysis.

### 3.2. XRD Analysis

Figure 3, Figure 4, Figure 5 and Figure 6 shows the phase assemblages of T20-OPC, T40-OPC, T20-CCGL30 and T40-CCGL30 hydrated from 1 to 28 d, respectively. As observed, the primary clinker phases of C_3_S and C_2_S are present throughout the hydration process in these four samples, and the main crystalline hydrates of ettringite (AFt) and portlandite (CH) are also precipitated in these samples at the beginning of 1 d (Figure 3) and exist during the whole curing age.

In order to better compare the influence of curing temperature on cement hydration, the overlapped C_3_S and C_2_S peak centered at 32.5° 2θ was selected for better comparison. As shown in Figure 3, compared with T20-OPC, the intensity of this peak decreased in T40-OPC, indicating that increasing the curing temperature accelerated cement hydration. Since CH in only precipitated from C_3_S and C_2_S hydration, the precipitation of CH was also enhanced, as evidenced by the intensity in the characteristic peak of CH centered at 18.0° 2θ also increasing accordingly. These changes were also observed at 3 and 7 d of curing (Figure 4 and Figure 5). However, the difference in the peak intensity centered at 32.5° 2θ is difficult to observe at 28 d (Figure 6), which reflected that increasing the curing temperature up to 40 °C is not effective in enhancing cement hydration.

A similar promotion effect in cement hydration with the increasing curing temperature can also observed in the samples of T20-CCGL30 and T40-CCGL30. Different from T20-OPC and T40-OPC, the phase of hemi-carboaluminate (Hc) with a characteristic peak centered at 32.5° 2θ was observed in T20-CCGL30 at 7 d due to the presence of limestone (Figure 5). After 28 d of curing, the characteristic peak of mono-carboaluminate (Mc) centered at 32.5° 2θ appeared while the peak of Hc was negligible. This resulted from the transformation from Hc to Mc. When the curing temperature increased to 40 °C, both the Hc and Mc peaks started to appear from 3 d. The Hc peak can still be observed at 7 d, but it disappeared at 28 d. This reflected that a higher curing temperature was conducive to not only the formation of Hc and Mc but also the transformation from Hc to Mc. Note that by the comparison of T20-CCGL30, the peak intensity of CH centered at 18.0° 2θ in T40-CCGL30 was also much lower from 3 d onwards, which is because the accelerated reaction of CCG under a higher reaction temperature consumed more CH.

### 3.3. Thermal Analysis

The TG/DTG curves of all samples at curing ages ranging from 1 to 28 d are illustrated in Figure 7, Figure 8, Figure 9 and Figure 10. From the DTG results, it can be seen that three major mass loss peaks are observed. The first peak of mass loss, occurring between 40 and 200 °C, primarily results from the dehydration process of calcium silicate hydrate (C-S-H) and ettringite (AFt). The second mass loss peak within the temperature range of 400–500 °C is attributed to the decomposition of portlandite (CH). Lastly, the mass loss occurring between 600 and 800 °C mainly arises from the decarbonation process of CaCO_3_. In addition, the mass losses in different temperature ranges were also calculated according to TG/DTG curves, and the results are listed in Table 3.

As observed in Figure 7, Figure 8 and Figure 9, the overall mass loss of T40-OPC was much higher than that of T20-OPC from 1 to 3 d, indicating the hydration of cement at the early curing ages was accelerated upon increasing the curing temperature. However, T40-OPC presented a comparable overall mass loss to T20-OPC at 7 d (Figure 10), but it was much lower at 28 d, indicating that the curing temperature has a limited or even negative influence on the hydration of cement upon further increasing the curing age from 7 to 28 d.

As shown in Figure 9, the overall mass loss of CCGL30 cured at 40 °C was still significantly higher than that cured at 20 °C at 7 d. On the contrary, the area of CH in T40-CCGL30 started to be much lower than that in T20-CCGL30 from 3 d, indicating that CH was greatly consumed during this period. It should also be pointed out that the sample of T40-CCGL30 exhibited traces of AFm-type hydrates at 3 d (see Figure 8), which reflected that CCG started to react with limestone to precipitate AFm-type hydration products such as Hc and Mc. These phases were only detected in T20-CCGL30 at 7 d or even later. This means that the reaction between CCG and limestone was promoted. These findings are consistent with the XRD observations, where increasing the curing temperature not only promoted cement hydration at early ages but also promoted the reaction of CCG, as evidenced by the accelerated consumption of CH and early appearance of AFm phases. The positive effect on the reactions of OPC and CCGL30 systems that resulted from increasing the curing temperature also well explained the compressive strength development of the corresponding mortars, as observed in Figure 2.

### 3.4. Pore Structure Analysis

The total porosity and pore size distribution of hardened pastes can be obtained based on the test results of MIP analysis. The effects of curing temperature on the pore structure of hardened pastes at 1 d and 28 d are illustrated in Figure 11 and Figure 12, respectively. The total porosity of hardened pastes is listed in Table 4.

At 1 d, the total porosity of hardened pastes decreased when the curing temperature was increased in both OPC and CCG systems (see Figure 11a). The increase in temperature promotes the hydration of the system, leading to the formation of more hydration products. As a result, the overall porosity of the system is reduced. The total porosity of the CCG system is higher than that of the OPC system at the same curing temperature, especially for the CCG system cured at 20 °C. This difference can be mainly attributed to the slower reaction kinetics of calcined coal gangue and limestone. Additionally, the total porosity of the CCG system cured at 40 °C is comparable to that of the OPC system cured at 20 °C, indicating that elevating the curing temperature significantly contributes to reducing the total porosity of the CCG system. The pore size distribution of hardened pastes at 1 d is illustrated in Figure 11b. As observed, the pores in hardened pastes can be categorized as gel pores (<0.01 μm), fine capillary pores (0.01–0.05 μm), medium capillary pores (0.05–0.1 μm), and large capillary pores (>0.1 μm). The refinement of the pore structure in the hardened paste is evidently observed with an increase in curing temperature, leading to a shift of the distribution curve from larger pores toward smaller ones.

As shown in Figure 12, unquestionably, the total porosity of hardened cement pastes decreases with the increase in curing age due to the continuous reaction of reactants that filled the pores. Similar to 1 d, the total porosity of the CCG systems decreased upon increasing the curing temperature at 28 d, indicating that a higher curing temperature continued to reduce the porosity of CCG. However, it is worth noting that in the OPC system, increasing the curing temperature leads to an increase in total porosity, which is possibly due to the adverse impact of high-temperature curing on the subsequent structural compaction. Figure 12b shows the pore size distribution of hardened pastes at 28 d. The pores of hardened pastes can also be categorized into four parts. The pore structure at 28 d is significantly finer compared to that of 1 d, which is mainly due to the continuous hydration process in the systems. The pore sizes of all systems are predominantly distributed within the range of fine capillary pores and medium capillary pores. The pore structure of OPC samples cured at 40 °C is more evenly distributed in medium capillary pores compared to that of OPC samples cured at 20 °C, which is consistent with the total porosity of the OPC sample cured at 40 °C being higher than that of the sample cured at 20 °C. The pore size distribution of CCG samples is predominantly concentrated in fine capillary pores. In contrast, the pore size distribution curve of CCG samples cured at 40 °C exhibits a greater shift toward gel pores. The CCG samples demonstrate a favorable pore structure in later stages, and elevating the curing temperature contributes to further refinement of the pore structure.

### 3.5. SEM Analysis

Figure 13 shows the morphology of T20-OPC, T40-OPC, T20-CCGLS30 and T40-CCGLS30 at 1 d. As shown in Figure 13a, T20-OPC presented loose morphology with many pores left by the free water, while denser morphology with less pores was available in T40-OPC (Figure 13b). Similarly, T40-CCGLS30 also revealed a much more compact structure than T20-CCGLS30. This can be mainly attributed to the increased curing temperature that greatly promoted cement hydration at 1 d, which produced more hydrates and densified the microstructure. These changes are also have significantly positive effects on the pore structures’ refinement (Figure 11) and early compressive strength development (Figure 2).

After 28 d of hydration (Figure 14), the continuous cement hydration precipitated more hydrates and forming a connected morphology of T20-OPC and T40-OPC. However, the morphologies of T20-OPC and T40-OPC are comparable, indicating that increased the curing temperature had a limited effect on the microstructure of OPC. On the contrary, the morphology of T40-CCGLS30 is still much denser than that of T20-CCGLS30 in this period, which is mainly due to the promotion effect of the increased curing temperature on CCG reactions that produced more hydrates (C-A-S-H gel, Hc and Mc). This also well explained the higher compressive strength of T40-CCGLS30 than that of T20-CCGLS30 at 28 d (Figure 2).

## 4. Conclusions

This research investigated the influence of curing temperature on the hydration properties of blended cement with limestone and calcined coal gangue. Based on the results and discussion, the following conclusions can be drawn:The compressive strength of the calcined coal gangue–limestone blended cements is effectively enhanced by elevating the curing temperature to 40 °C, particularly in low clinker systems which exhibit greater sensitivity toward temperature.In calcined coal gangue–limestone blended cements, increasing the curing temperature not only promotes cement hydration but also facilitates the reaction of CCG, which precipitated more hydrates such as C-A-S-H gel, Hc and Mc.The total porosity of the calcined coal gangue–limestone blended cements decreases with increasing curing temperature. At 20 °C, the pore size distribution is mainly concentrated in fine capillary pores. At 40 °C, there is a more pronounced shift toward gel pores observed in the pore size distribution curve.The calcined coal gangue–limestone blended cements will exhibit a denser microscopic morphology when the curing temperature is increased.

## Figures and Tables

**Figure 1 materials-17-01721-f001:**
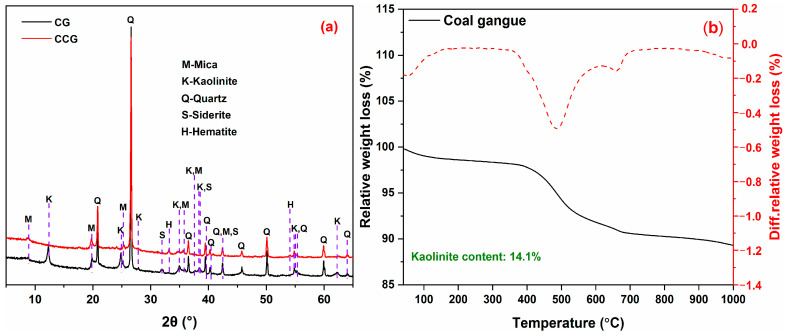
(**a**) XRD patterns and (**b**) TG/DTG curves of CG and CCG.

**Figure 2 materials-17-01721-f002:**
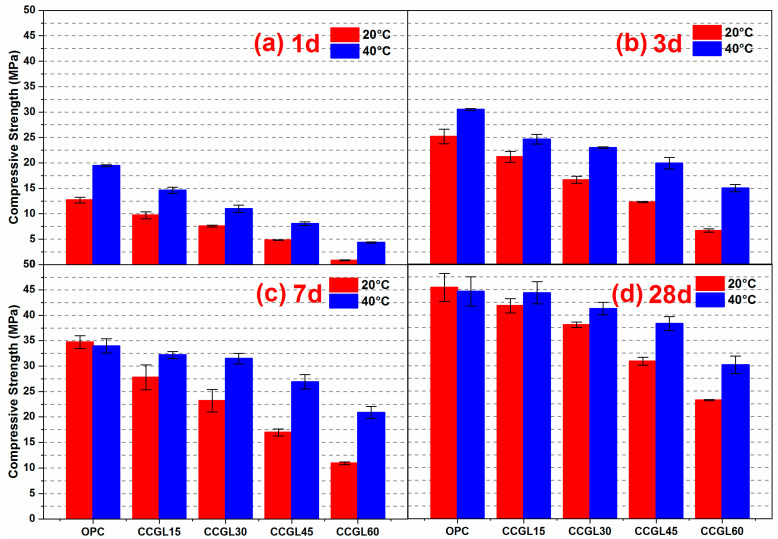
Compressive strength of mortar samples cured at 20 and 40 °C up to 28 d.

**Figure 3 materials-17-01721-f003:**
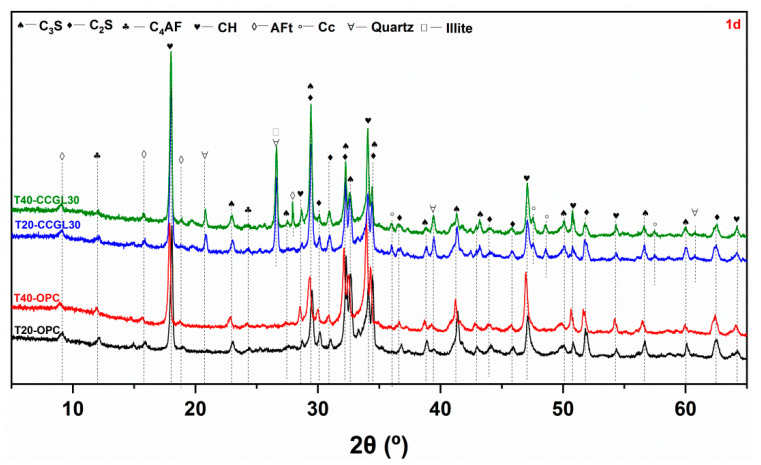
XRD patterns of paste samples cured at 20 and 40 °C at 1 d (C_3_S: tricalcium silicate, C_2_S: dicalcium silicate, C_4_AF: ferrite, CH: portlandite, AFt: ettringite, Cc: calcite).

**Figure 4 materials-17-01721-f004:**
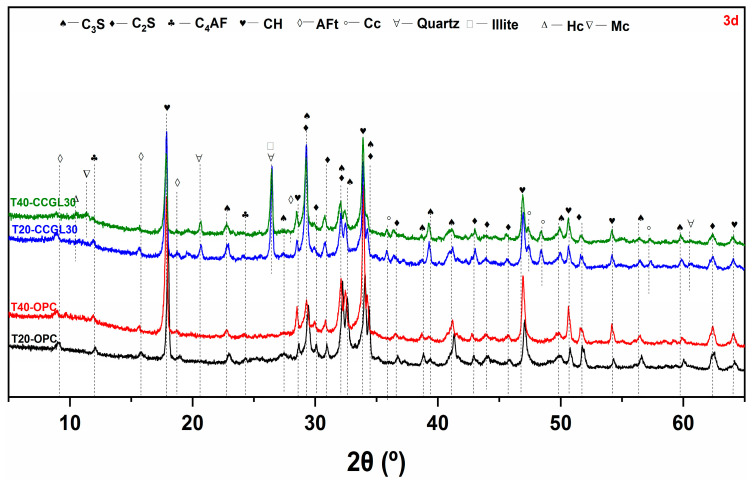
XRD patterns of paste samples cured at 20 and 40 °C at 3 d (C_3_S: tricalcium silicate, C_2_S: dicalcium silicate, C_4_AF: ferrite, CH: portlandite, AFt: ettringite, Cc: calcite).

**Figure 5 materials-17-01721-f005:**
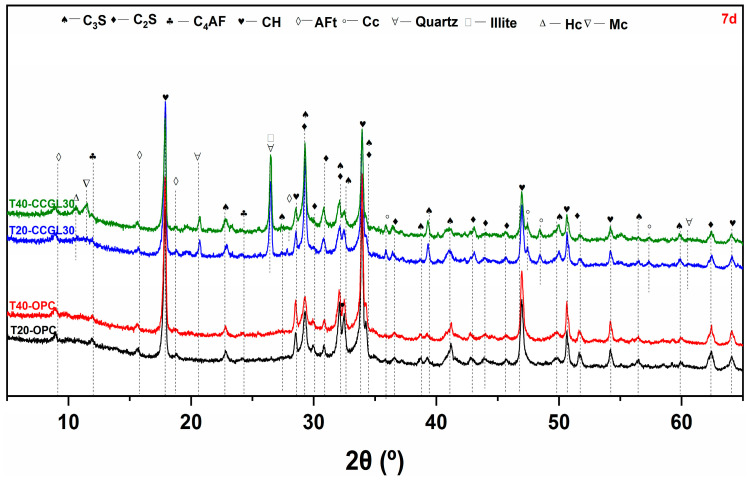
XRD patterns of paste samples cured at 20 and 40 °C at 7 d (C_3_S: tricalcium silicate, C_2_S: dicalcium silicate, C_4_AF: ferrite, CH: portlandite, AFt: ettringite, Cc: calcite).

**Figure 6 materials-17-01721-f006:**
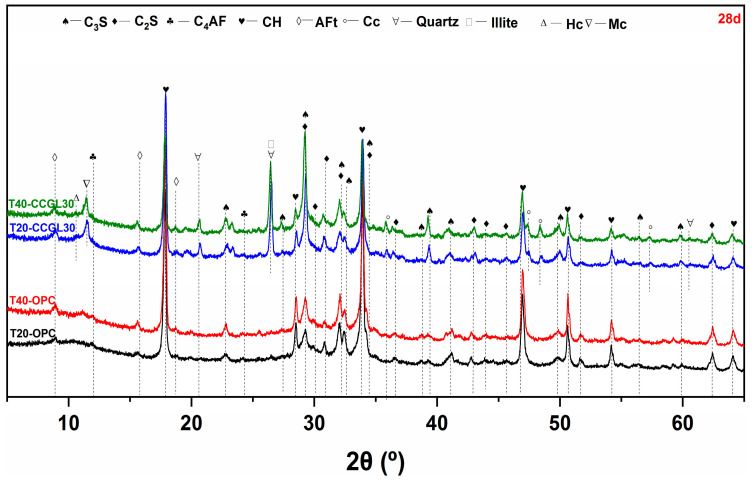
XRD patterns of paste samples cured at 20 and 40 °C at 28 d (C_3_S: tricalcium silicate, C_2_S: dicalcium silicate, C_4_AF: ferrite, CH: portlandite, AFt: ettringite, Cc: calcite).

**Figure 7 materials-17-01721-f007:**
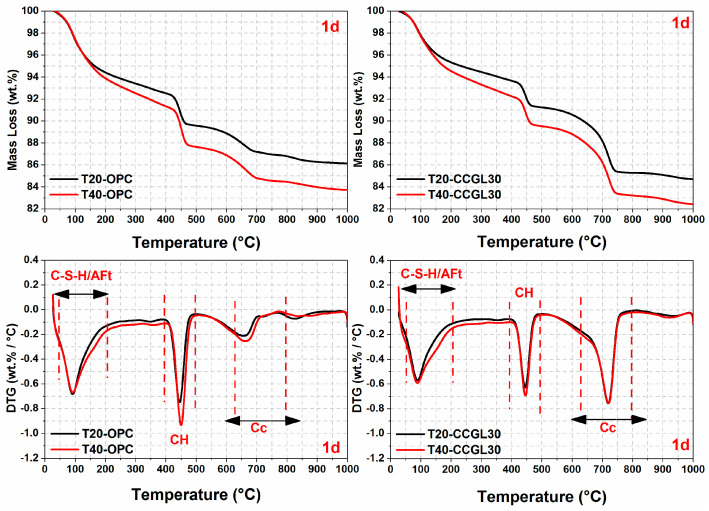
TG-DTG curves of hardened pastes at 1 day.

**Figure 8 materials-17-01721-f008:**
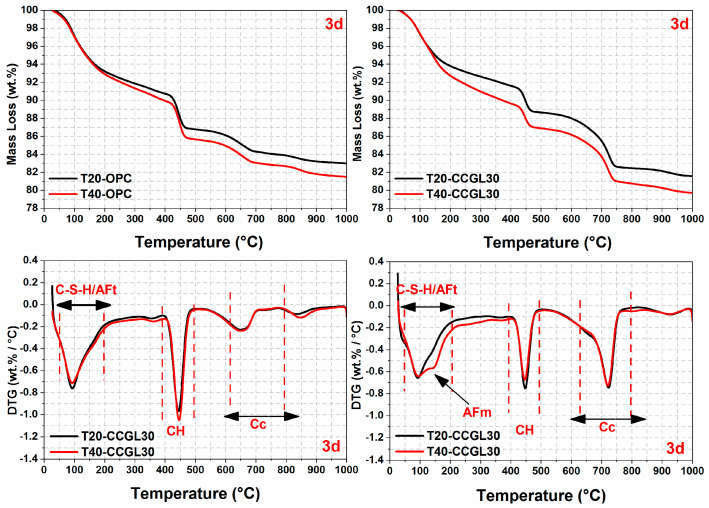
TG-DTG curves of hardened pastes at 3 days.

**Figure 9 materials-17-01721-f009:**
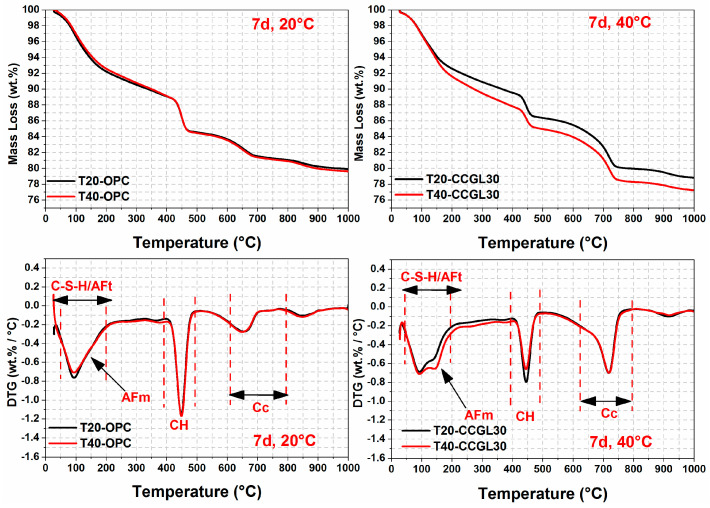
TG-DTG curves of hardened pastes at 7 days.

**Figure 10 materials-17-01721-f010:**
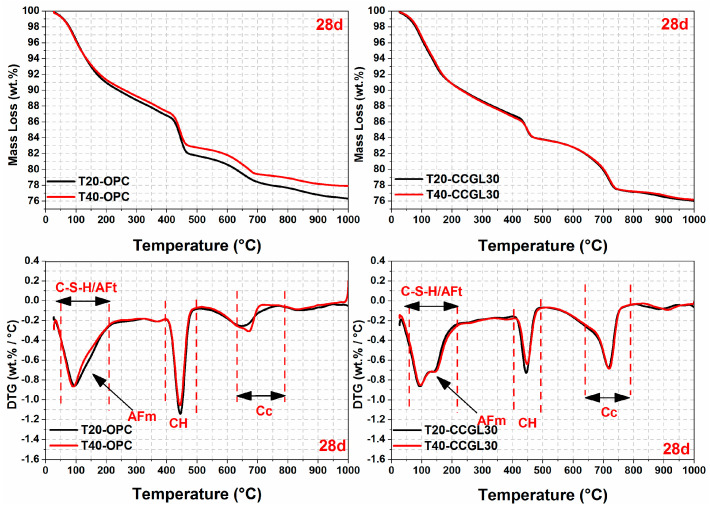
TG-DTG curves of hardened pastes at 28 days.

**Figure 11 materials-17-01721-f011:**
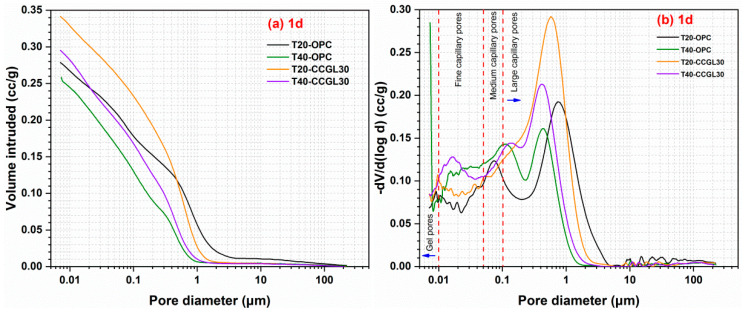
(**a**) Cumulative and (**b**) incremental pore volume of hardened pastes at 1 d.

**Figure 12 materials-17-01721-f012:**
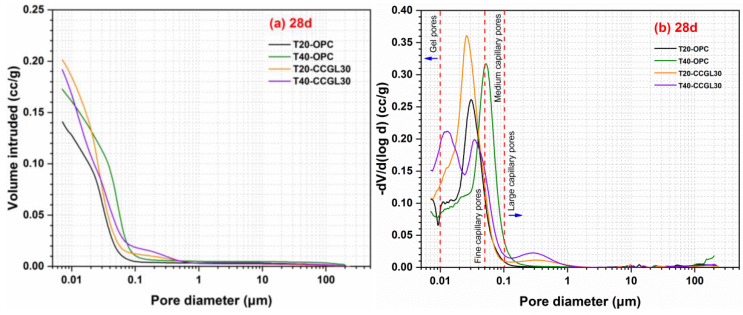
(**a**) Cumulative and (**b**) incremental pore volume of hardened pastes at 28 days.

**Figure 13 materials-17-01721-f013:**
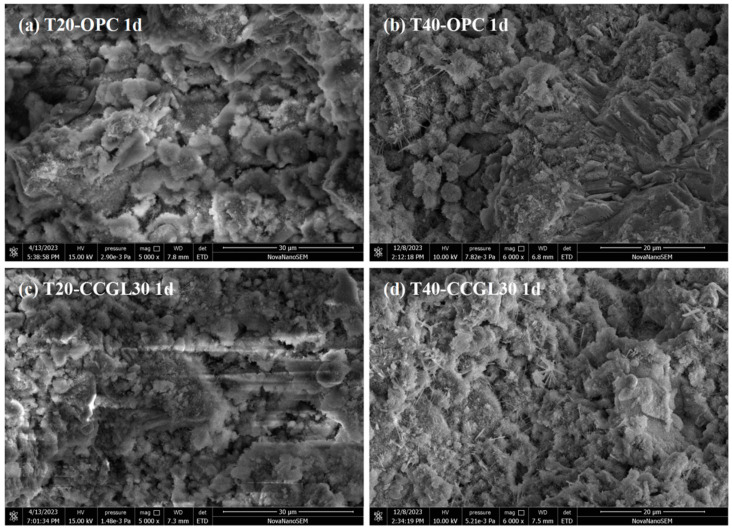
SEM observation of hardened pastes at 1 d.

**Figure 14 materials-17-01721-f014:**
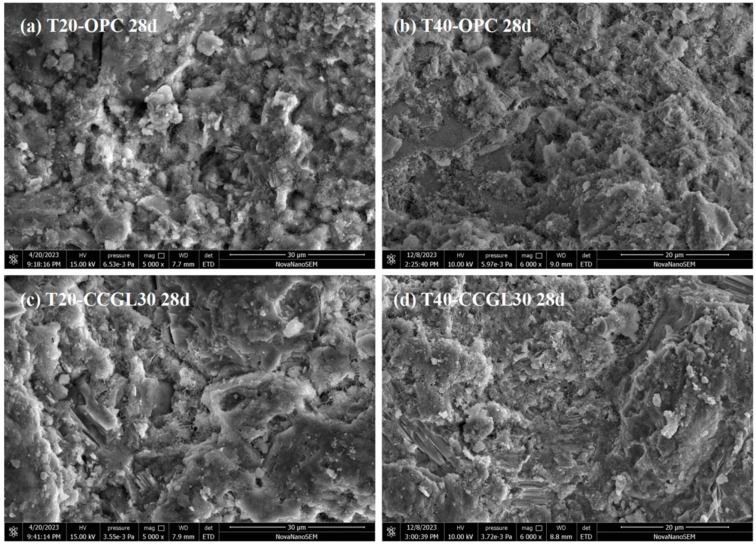
SEM observation of hardened pastes at 28 d.

**Table 1 materials-17-01721-t001:** Chemical compositions of raw materials (wt.%).

	CaO	SiO_2_	Al_2_O_3_	Fe_2_O_3_	MgO	SO_3_	Na_2_O	K_2_O	TiO_2_	P_2_O_5_	LOI
Cement	54.0	17.6	3.7	3.3	2.4	2.9	0.1	0.7	0.3	0.1	14.6
Limestone	55.6	5.2	2.4	1.3	2.7	0.2	0.03	0.6	0.1	0.3	31.6
CG	1.2	43.6	19.3	2.6	0.4	0.1	0.3	1.1	0.7	0.1	16.5

**Table 2 materials-17-01721-t002:** Detailed mix proportions of the mortar (g, 100 g base for binder).

T	Notation	OPC	CCG	LS	Gypsum	Water	Sand
20 °C	T20-OPC	100	0	0	0	50	300
20 °C	T20CCGL15	84.8	10	5	0.2	50	300
20 °C	T20CCGL30	69.6	20	10	0.4	50	300
20 °C	T20CCGL45	54.4	30	15	0.6	50	300
20 °C	T20CCGL60	39.2	40	20	0.8	50	300
40 °C	T40-OPC	100	0	0	0	50	300
40 °C	T40-CCGL15	84.8	10	5	0.2	50	300
40 °C	T40-CCGL30	69.6	20	10	0.4	50	300
40 °C	T40-CCGL45	54.4	30	15	0.6	50	300
40 °C	T40-CCGL60	39.2	40	20	0.8	50	300

**Table 3 materials-17-01721-t003:** Relative mass loss at different temperature ranges (%).

Curing Age	Stage	OPC		CCGL30	
		20 °C	40 °C	20 °C	40 °C
1 d	40–200 °C	5.41	5.82	4.89	5.93
	400–500 °C	2.97	3.74	2.47	2.79
	40–1000 °C	9.69	11.17	13.9	15.81
3 d	40–200 °C	6.63	6.59	6.58	7.82
	400–500 °C	3.69	4.23	3.04	4.61
	40–1000 °C	11.91	12.17	16.69	17.08
7 d	40–200 °C	6.84	6.23	7.66	8.78
	400–500 °C	4.58	4.73	3.26	2.96
	40–1000 °C	13.13	12.05	18.49	19.48
28 d	40–200 °C	7.94	8.01	9.65	9.82
	400–500 °C	4.42	4.59	3.18	3.01
	40–1000 °C	14.21	13.85	20.62	20.44

**Table 4 materials-17-01721-t004:** Total porosity of samples (%).

	1 d	28 d
T20-OPC	27.91	14.07
T40-OPC	25.88	19.17
T20-CCGL30	34.18	20.10
T40-CCGL30	29.47	19.14

## Data Availability

The data used in this study can be required from the corresponding author. The data are not publicly available due to information that could compromise the research participants’ privacy.

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
