# Peer review of "Influence of Curing Temperature on the Performance of Calcined Coal Gangue–Limestone Blended Cements"

_materials, 2024, doi:10.3390/ma17081721_

Round 1
Reviewer 1 Report
Comments and Suggestions for Authors
The authors in the present manuscript show that the utilization of calcined coal gangue (CCG) and limestone for the preparation of blended cement is an efficient approach to address the issue of coal gangue disposal. However, the compressive strength development of blended cement is slow, particularly at high substitution levels of CCG. Therefore, this study aimed to promote the hydration and mechanical properties of the calcined coal gangue-limestone blended cements by increasing the curing temperature. In this study, the samples were cured at two different temperatures, namely 20 and 40 °C. Four groups of samples with 15 wt.%, 30 wt.%, 45 wt.% and 60 wt.% cement substitutions using CCG and limestone (2:1 mass ratio). The compressive strength, hydration and microstructure were investigated at the ages from 1 d to 28 d. X-ray diffraction (XRD) and thermogravimetry (TG) were used to study the hydration behavior of samples. Mercury intrusion porosimetry (MIP) and scanning electron microscopy (SEM) were used to determine the microstructure of samples. The results indicate that an increase in curing temperature significantly promotes the compressive strength of the calcined coal gangue-limestone blended cements from 1 to 28 d. The authors should address the following issues and information’s before publication acceptance in the prestigious ‘Materials’ Journal:
1. In Introduction, authors should add a Table that compares the different coal, cure temperature preparation methods, cement weight ratio, and properties with published literatures.
2. In Materials and methods, authors should explain on what basis selected cure temperatures, i.e. 20 and 40 °C.
3. What is the carbon content in the coal gangue and what is the role of carbon in this study?
4. In Figure 2, authors should keep the same y-axis scale for better comparation of compressive strength of mortar samples.
5. In Figure 11 and 12, authors should explain why the pore diameter of the samples reduced after 28 days?
6. In Introduction section, authors should discuss bit more about coal reserves for other countries as well. Authors may go through this publication for more details and cite accordingly: https://doi.org/10.1016/j.carbon.2023.118447

Comments on the Quality of English LanguageMinor editing of English language required.
Reviewer 2 Report
Comments and Suggestions for Authors
This paper aims to evaluate two curing temperatures for four replacement levels of cement using calcined coal gangue and limestone. The manuscript is well-written and structured. Some recommendations are presented below:
1. Figure 2. It is better to compare the figures if you use the same y-values for all the ages.
2. Figures 7-9. Can the lower DTG values for OPC systems attributed to carbonation be associated with different polymorphic carbonates? If yes, the DTG area for carbonates needs to be revised.
3. SEM analyses are always complicated to evaluate. Other analyses (CT, helium pycnometer, etc.) could be better used to assess porous materials.
4. How are you sure that C-A-S-H gel is formed instead of C-S-H gel formed?
5. MIP are exciting results. Can you present the values of total porosity in a table?
6. Can you calculate the amount of CH and Cc present in the samples through TG-DTG? This could better explain the gains of using this material as a replacement for Portland cement.
Comments on the Quality of English LanguageAfter revision, this paper can be accepted for publication.
Reviewer 3 Report
Comments and Suggestions for Authors
The author tried to show people the impact of curing temp on a few composites’ properties. However, the writing is not so good, and the entire draft is formed by simply stacking a lot of data and lengthy analysis. I would suggest major revision based on the current version.
1. I didn’t see the significance of this work stated clearly. Instead of saying’ investigating the effect of curing temperature on hydration and microstructure properties of …. (line107)’, the author should focus on showcase people why you wanted to understand this phenomenon. Based on a better understanding of it, what’s potential problems or bottlenecks that can be solved.
2. Why are these composites studied in your work (table 2)? any practical use that you chose these recipes?
3. In the introduction part, can author provide more background on hydration kinetic model of this specific material/composite? Can the curing process be predicted from some simple kinetic model?
4. Fig.3 to 6, need more annotations (or arrows) for the peaks that authors have discussed in the main text.
5. It’s not convenient to compare curves in different figures (e.g. fig7 to 10). Better to think about how to visualize this better and move some unnecessary data to supplemental information.
6. line371: based instead of ‘besed’
Comments on the Quality of English Languagena
Round 2
Reviewer 3 Report
Comments and Suggestions for Authors
The paper can be accepted in the current form